# Fast Jukebox: Accelerating Music Generation with Knowledge Distillation

**Michel Pezzat-Morales [1], Hector Perez-Meana [1,*] and Toru Nakashika [2]**

[1] Graduate and Research Section, Mechanical and Electrical Engineering School Instituto Politécnico Nacional, Mexico City 04440, Mexico

[2] Department of Information System Fundamentals, Graduate School of Information Systems, The University of Electro-Communications, Tokyo 182-8585, Japan

**\*** Correspondence: hmperezm@ipn.mx

**Featured Application: This paper presents an improvement of the Jukebox system for music generation, which significantly reduces the inference time.**

**Abstract:** The Jukebox model can generate high-diversity music within a single system, which is achieved by using a hierarchical VQ-VAE architecture to compress audio in a discrete space at different compression levels. Even though the results are impressive, the inference stage is tremendously slow. To address this issue, we propose a Fast Jukebox, which uses different knowledge distillation strategies to reduce the number of parameters of the prior model for compressed space. Since the Jukebox has shown highly diverse audio generation capabilities, we used a simple compilation of songs for experimental purposes. Evaluation results obtained using emotional valence show that the proposed approach achieved a tendency towards actively pleasant, thus reducing inference time for all VQ-VAE levels without compromising quality.

**Keywords:** music generation; autoregressive prediction; knowledge distillation; VQ-VAE

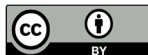

## 1. Introduction

Music has been one of the most representative features of any culture around the world since ancient times. Because of this, many researchers have established that the relationship existing between the music score and real music is quite similar to that existing between written text and real speech [1]. Here, we can describe music as consisting of two levels: the music score, which is a symbolic and highly abstract expression, and the sound, which is a continuous and concrete signal that provides the details that we can hear [1]. Thus, music generation can be divided into three stages: in the first stage, the composer generates the music scores; in the second stage, the musician or singer generates the performance using the scores; and finally, in the last stage, the performance results in a musical sound by adding different instruments, which are perceived by the listeners [1]. From the above, the automatic music generation can be divided into three levels: (a) score generation, which covers polyphonic music generation, accompaniment generation, interactive generation, etc.; (b) performance generation, which includes rendering the performance, which does not change the stablished score features, and the composed performance that models both the music score and performance features; and (c) audio generation, which pays attention to the acoustic information using either the waveform or spectral approaches [1]. Finally, by adding lyrics to the score, proving it with timbre and style, it is possible to realize singing synthesis.

Automatic music generation has been a topic of active research during the last 60 years, during which many methods have been proposed, including grammar rules, probabilistic models, evolutionary computing, neural networks, etc. [2]. More specifically,

thanks to advancements in the field of deep learning being applied in computer vision [3] and natural language processing [4,5], different proposals have been made to create artificial speech or sounds, with models such as recurrent neural networks [6], generative adversarial networks [7,8], variational autoencoders [9], and transformers [10]. One of the most recent and successful contributions in this field is the Jukebox [11,12]. The Jukebox uses VQ-VAE [13], an approach that compresses and quantizes extremely long context inputs in the raw audio domain into shorter-length discrete latent encoding using a vector quantization approach. After training the VQ-VAE, a prior is learned over the compressed space to generate new audio samples. Generating discrete codes not only allows audio conditioning, but also creates music from different genres and instruments, including singing voices. This system provides impressive results; however, the inference stage is very low and tremendously slow. To tackle this problem, a Fast Jukebox is proposed in this paper to reduce the inference time, in which we distilled the prior learning process by comparing a large autoregressive model against a smaller one. With training losses borrowed from those recently proposed by Tiny Bert [14], this proposal generates audio several times faster than smaller Jukebox architectures.

The paper is organized as follows: Section 2 provides a description of some related works, and Section 3 presents the background required in the presented work. The methodology used to develop the proposed scheme is provided in Section 4. The evaluation results are provided in Section 5. Section 6 provides a discussion about the proposed research. Finally, the conclusions of this research are provided in Section 6.

## 2. Related Work

Due to the difficulty represented by modeling raw audio, most generative models use a symbolic approach. Some examples are MidiNet [15] and MusGAN [16], which use generative adversarial networks, as well as MusicVAE [17] and HRNN [18], which are based on hierarchical recurrent networks. The main drawback of the symbolic approach is that it constrains the generated music to a specific sequence of notes and a fixed set of instruments. This constraint has motivated researchers to investigate non-symbolic approaches. WaveNet [19] performs autoregressive sample-by-sample probabilistic modeling of the raw waveform using a series of dilated convolutions to exponentially increase the context length. The parallel use of Wavenet [20] and Clarinet allows [21,22] faster sampling from a continuous probability distribution distilled using a pre-trained autoregressive model. Parallel WaveGAN and MelGAN [23,24] are GAN-based approaches that directly model audio waveforms, achieving a similar quality to WaveNet models, with significantly fewer parameters. Transformers [25,26] have previously been introduced in the context of neural machine translation. Eventually, this encompassed music synthesis and composition with the likes of Music Transformer [27,28] to autoregressively predict MIDI note events. Due to the context limitations of the original model, authors such as Child et al. [29] introduced the sparse factorization of the attention matrix to reduce the self-attention complexity from $O(N\sqrt{N}d)$ to $O(Nd^2)$, where N is the sequence length and d is the transformer dimension. The autoregressive generation of discrete audio tokens has been studied recently in works such as those reported in [30,31], not only because it allows the processing of long-term structures in the raw audio domain, but it also serves to condition audio with other embedded information, such as text and lyrics. In our research, we trained the model not only with maximum likelihood, but also using knowledge distillation [32]. The latter allowed us to obtain the same quality of Jukebox model without the need to add trainable parameters. By taking a previously large prior model to serve the role of the teacher, a smaller model takes the role of the student during training. The forked code and samples are shared for further research and the evaluation process [33,34].

## 3. Background

According to [35], the Jukebox forms a part of embedding approach models. This is because the audio for music is usually sampled at 44KHz, with 16 bits of precision for each sample. This means that for one minute, there are more than 5 million instances of 16 bits of information. This is why reducing dimensionality in sound processing can sometimes be convenient, even if it is not always necessary. A VQ-VAE (vector-quantized variational autoencoder) is applied to obtain embedded vectors from an input sequence $x = \langle x_t \rangle_t^T$. By encoding and quantizing audio samples to a sequence of K possible values, that are obtained for $z = \langle z_s \epsilon K \rangle_s^S$. The token $z_s$ is the latent discrete representation over which the prior is trained hierarchically to form a different temporal resolution. The model is illustrated in Figure 1.

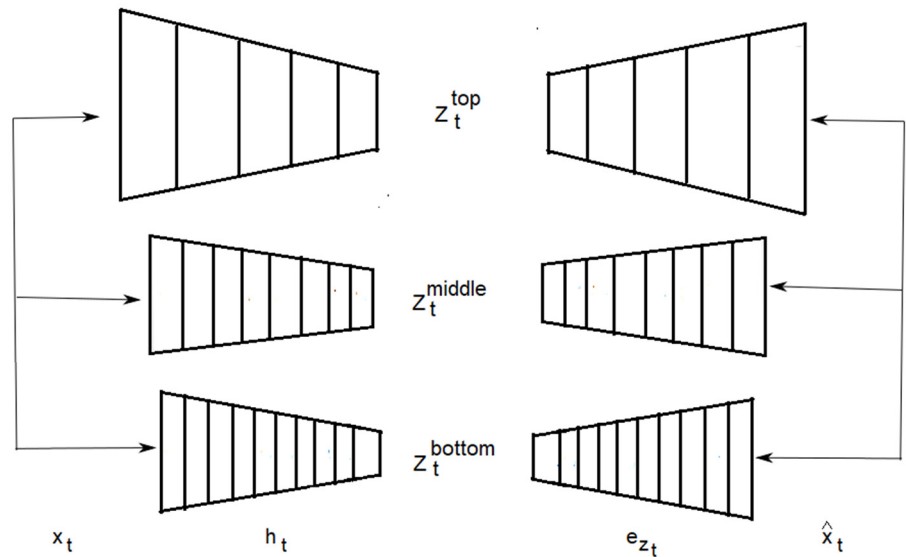

**Figure 1.** Hierarchical VQ-VAE of three levels of abstraction. Each level is a model with separate priors conditioned on upper levels. $h_t$ is the latent vector obtained from the encoder, and $e_{z_t}$ is its nearest vector from codebook C= $\langle e_k \rangle_{k=1}^K$.

## 4. Methodology

At each level of encoding, an autoregressive model was trained to learn the prior over the compressed spaces. This process resulted in the following:

$$p(z) = p(z^{top}, z^{middle}, z^{bottom}) \tag{1}$$

$$p(z) = p(z^{top})p(z^{middle}|z^{top}), p(z^{bottom}|z^{top}, z^{middle}) \tag{2}$$

where $p(z^{top})$ is the top-level prior, while $p(z^{middle}|z^{top})$ and $p(z^{bottom}|z^{top}, z^{middle})$ are the up samplers. Each model level was trained separately. Sparse transformers were proposed to solve this autoregressive modeling problem. The sparsity patterns are shown in Figure 2. Due to the computational expensive nature, knowledge distillation was used to accelerate the sampling time.

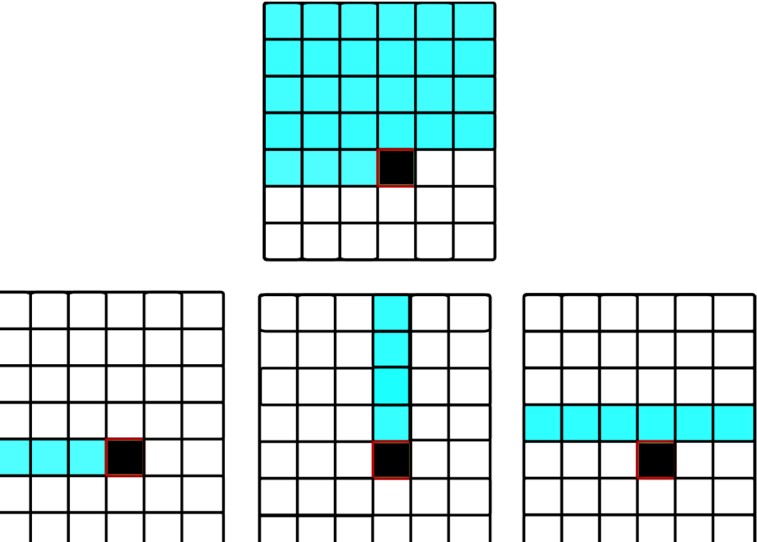

**Figure 2.** Attention patterns used in the sparse transformer. The first row shows the full attention of the vanilla transformer, since it is autoregressive all squares are marked up until the last iteration input (dark square). The second row shows the attention patterns of row attention, column attention, and previous row attention from left to right, respectively.

*Knowledge Distillation*

Knowledge distillation [2] is a method proposed to reduce size and computation costs during the inference stage in different models, such as transformers. The idea was to produce a large computational model to transfer its information to a smaller one. Once the large teacher model had been trained, its information would be used to train the student network so it could mimic its behavior. The most intuitive method of carrying out knowledge distillation was proposed by Hinton et al. [32]; here, the loss function was defined with cross entropy between the teacher and the student at predicted logits of $z^T$ and $z^s$, respectively.

$$L_{ce} = \sum_i z_i^T log(z_i^s)$$
(3)

In addition, we applied attention-based distillation and hidden-state-based distillation, as in [13], to strength transfer the information between the teacher and student using learning intermediate structures:

$$L_{hidn} = MSE(H^T, H^S)$$
(4)

where $H^T \in R^{l \times d}$ and $H^S \in R^{l \times d}$ represent the values shown in Figure 3, which are matrices with a sequence length l and transformer dimension d. The MSE evaluation was carried out with attention matrices $A^S$ and $A^T$, also illustrated in Figure 3. The Jukebox used a sparse-attention transformer with a different attention pattern for each layer. To keep the same level of attention for the student, the correspondence between layers was carried out as shown in Figure 4. Each layer's attention pattern was grouped together and distilled the information to the group in the student transformer. The layer proportion depended on the size of both the student and teacher transformers.

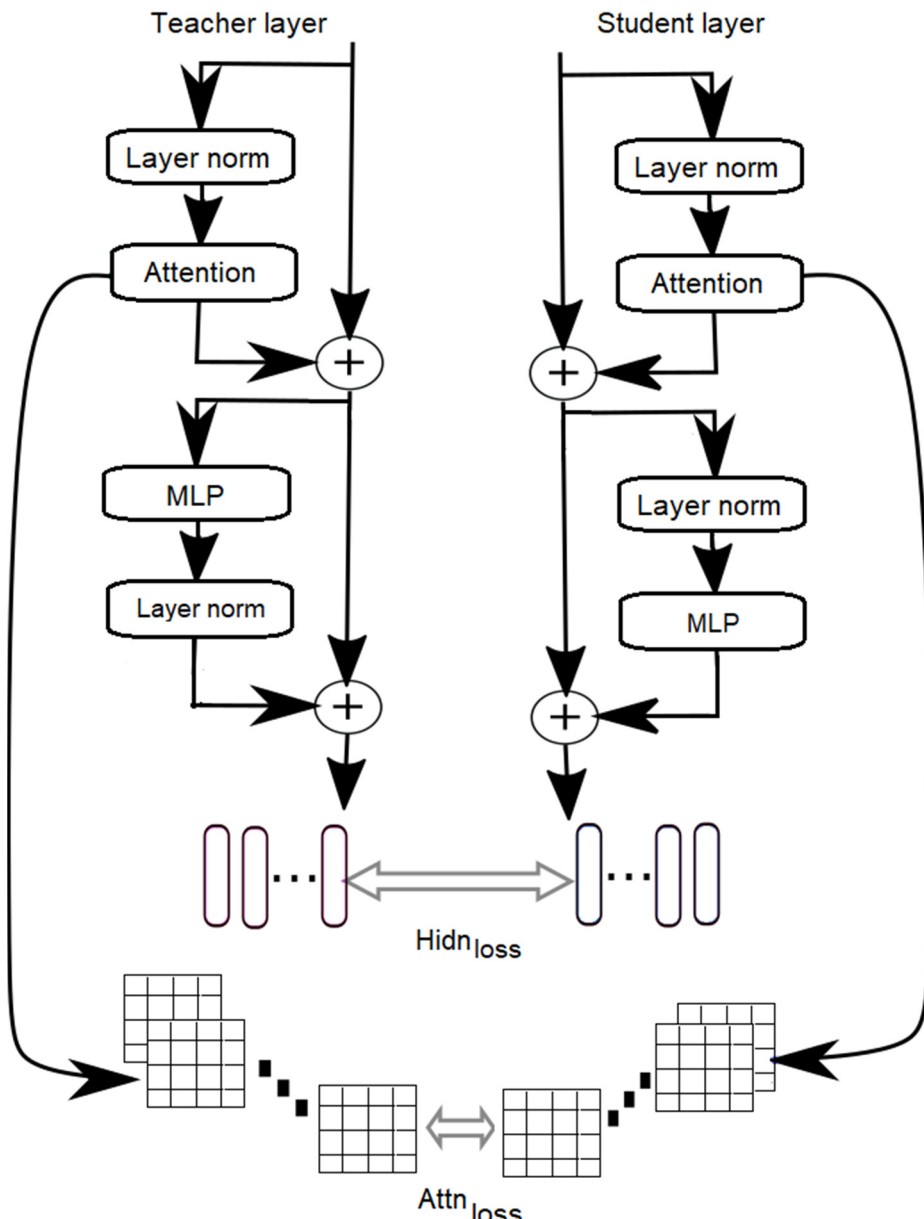

**Figure 3.** Transformer layer distillation for the prior Jukebox model architecture. Which is the typical autoregressive Transformer structure. Teacher and student outputs are compared in the attention block residual stage and outputs of the layers.

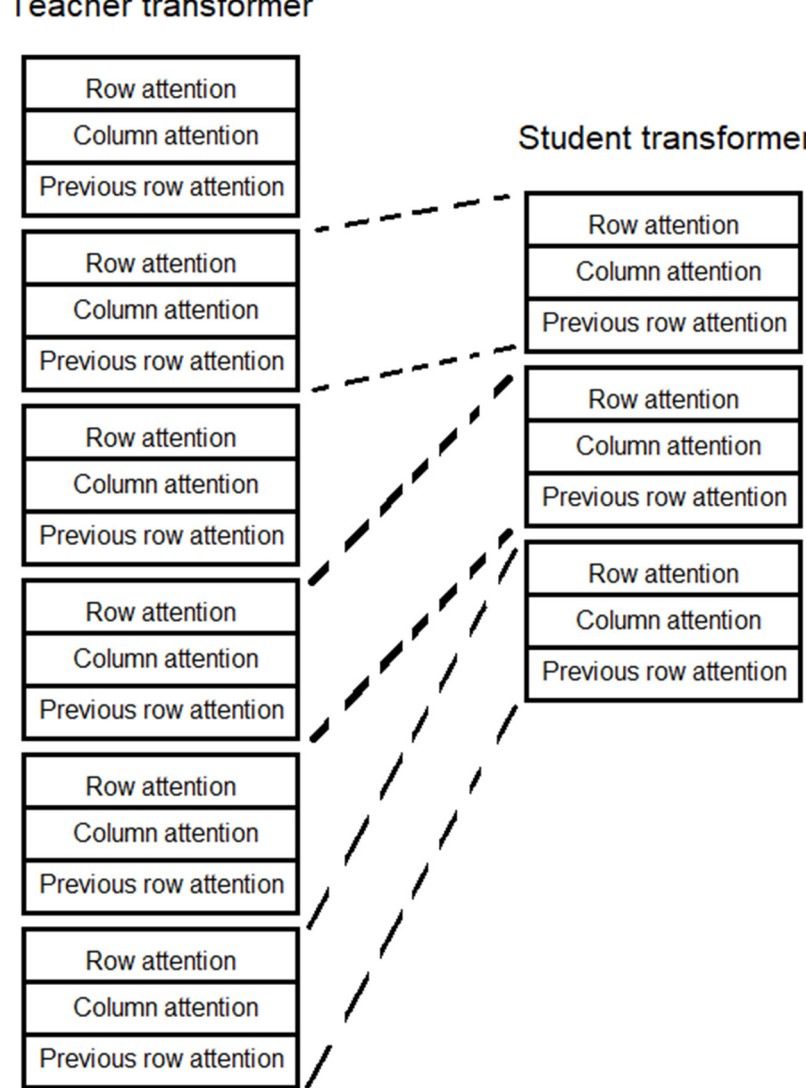

**Figure 4.** Layer correspondence based on attention pattern. Each attention pattern (row attention, column attention, and previous row attention) from the teacher model was consecutive.

## 5. Results

For the experiments, the compressed codes were obtained from the previously trained VQ-VAE model from the original jukebox code repository. It was a three-level hierarchical VQ-VAE, and its architecture is detailed in the original paper [11]. Audio was compressed to 128× of its input dimension on the top level, and to 32× and 8× on the middle and bottom levels, respectively. Since in this paper, our focus relies on reducing inference time rather than showing music diversity, we chose the training data to be much smaller than the one used in the original Jukebox proposal. The training data consisted of a compilation of songs by a single commercial music interpreter. This encompassed around 300 32-bit-resolution songs sampled at 44.1 KHz. It encoded the audio by taking 18, 6, and 1.5 s of raw audio during training from the top, middle, and bottom levels, respectively. For the sake of simplicity, we trained the top prior unconditionally, without lyrics or labels, and adopted almost the same architecture as in the 1-billion-parameter example, which encompassed the three-factored attention-sparse functions, whose patterns are shown in Figure 4 (further details can be seen in the Jukebox paper

[11]). The teacher model consisted of 72 layers of sparse attention for all prior and up samplers. Two student models were trained; they had 24 layers and 18 layers, respectively. The models were trained using Equation (4) for 20 epochs, and then using Equation (3) for a further 50 epochs.

### 5.1. Samples

Table 1 shows all ablation studies undertaken in this work, along with the sampling times. The samples consisted of 60 s of audio obtained in a single GPU. All the variations from Table 1 have been shared for listening [24]. The samples show that even though the audio quality was far from ideal, it moved closer to the teacher-generated audio. When compared with the samples generated from the same student architecture without distillation, the distillation process enhanced the positive characteristics of the teacher baseline. Figure 5 shows the waveform representation of 5 s of audio for all the models trained; it follows that knowledge distillation [2] makes the signal more closely resemble the complexity of both the teacher and dataset audio. In Figure 6, the level of abstraction for both the teacher and student, and the capability of detailing audio, are illustrated by a mel-spectrogram representation of the samples.

**Table 1.** List of samples shared for this research. All of them consisted of 60 s of audio obtained with a GPU Tesla P100.

| System | Speedup |
| --- | --- |
| Jukebox Base (Teacher) | 1.0× |
| Jukebox 24 | 3.0× |
| Jukebox 18 | 4.0× |
| Jukebox 24 (Student) w/o Pred | 3.0× |
| Jukebox 18 (Student) w/o Pred | 4.0× |
| Jukebox 24 (Student) | 3.0× |
| Jukebox 18 (Student) | 4.0× |

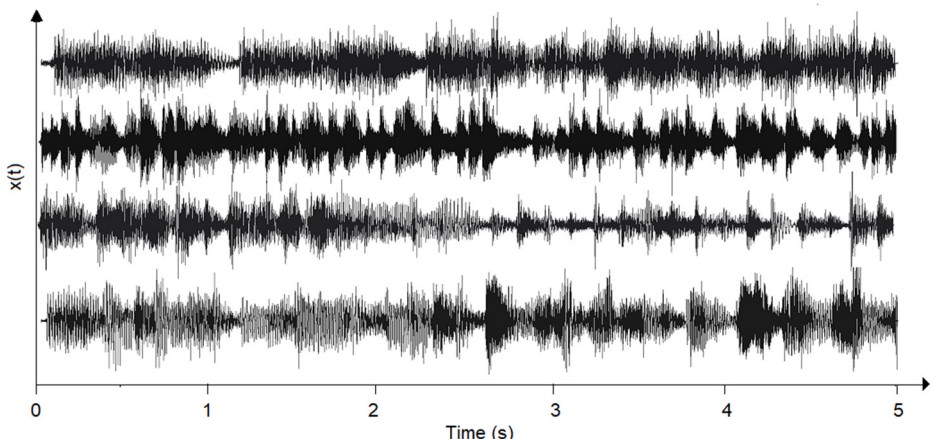

**Figure 5.** Example of the dataset waveform compared to samples obtained from both teacher and student models, each lasting 5 s. All the audio was normalized within a range of −1 to 1. The first row shows an example of the dataset used, the second row is the teacher transformer, and the third and fourth rows correspond to student models of 24 and 18 layers, respectively.

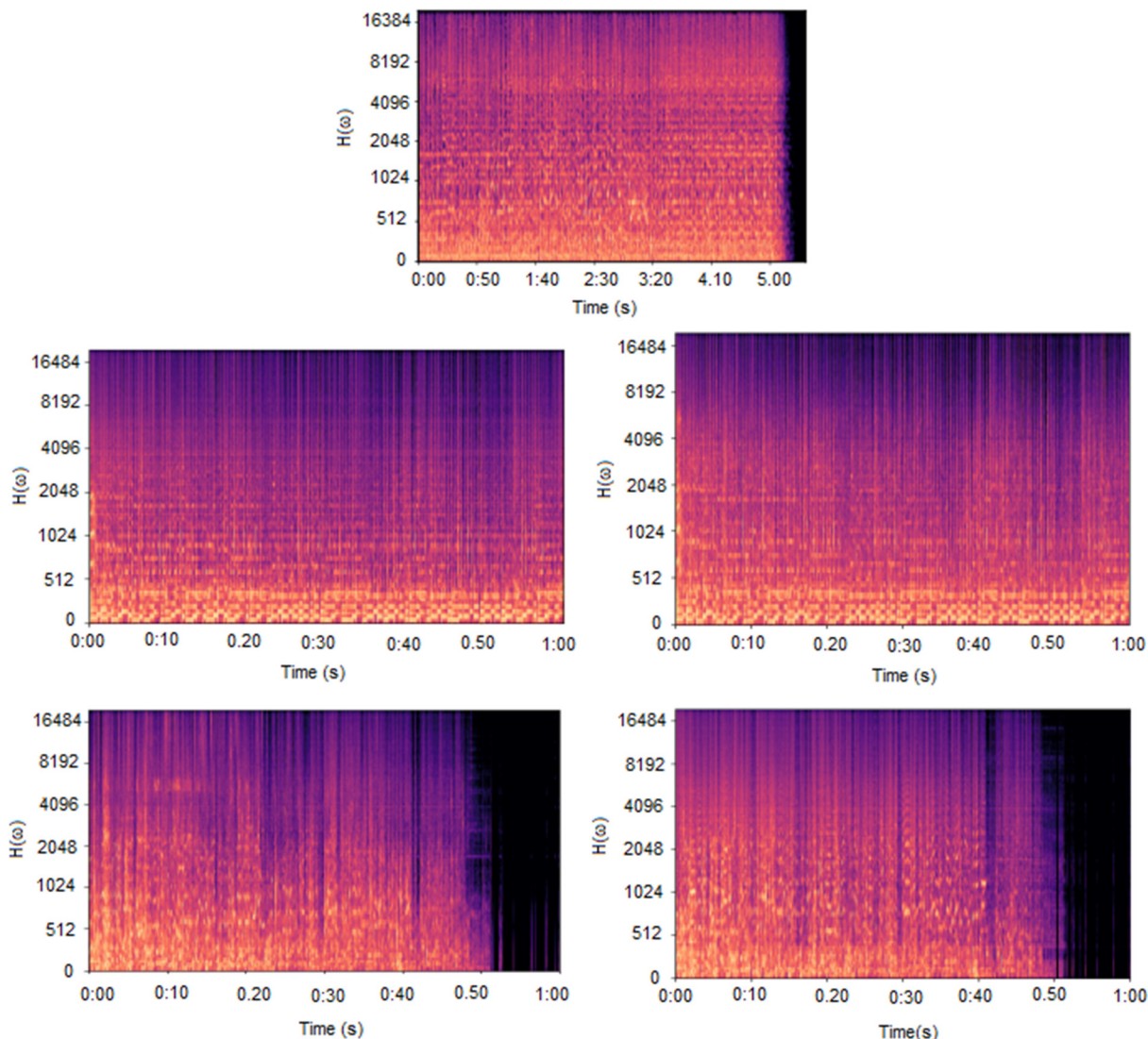

**Figure 6.** Mel-spectrum of the audio signal. On the top is a sample from the music dataset. The samples in the middle are from the top prior and up sample teacher model, left and right, respectively. The bottom samples are the student model, top prior and up-sampled. The up-sampling stage made the time resolution more detailed and reduced the low-frequency noise.

*5.2. Emotional Valence Evaluation*

Performing a quantitative evaluation of music is difficult [36]. The reason is that music itself is subjective in terms of quality. Despite this, it has been a subject of study by many researchers. Thus, the samples were analyzed using the classifier proposed by Sukhavasi et al. [37], using the same labels of emotional valence made by Godwin et al. [38] in his deep music evaluation analysis. They consist of four labels describing the combination of valence and arousal, where valence refers to the pleasure provoked by an external agent in a person, while arousal is related to the amusement of this agent. Therefore, the labels are 'activated pleasant' (AP), 'activated unpleasant' (AU), 'deactivated pleasant' (DP), and 'deactivated unpleasant' (DU). Together, they describe the arousal and valence responses of an individual when listening to music. The classifier was trained with these four labels, as in the original work. After 180 epochs, it reached an area under the precision–recall curve (PR-AUC) of 0.5046 and an area under the receiver operator characteristics curve (ROC-AUC) of 0.7537. Table 2 shows that real music is considered to be in the activated pleasant (AP) category, which was expected

from commercial music. Deep generated music was spread more evenly across all labels, with a tendency to be actively unpleasant. Our proposed distilled version achieved a categorization inclined slightly more towards actively pleasant.

**Table 2.** Mean of the classification of samples with emotional labels. Activated pleasant (AP), activated unpleasant (AU), deactivated pleasant (DP), and deactivated unpleasant (DU). More than a hundred samples were classified. Generative methods were expected to perform similarly to real music but were spread across the labels more evenly.

| Samples | AP | DP | AU | DU |
| --- | --- | --- | --- | --- |
| Real Music | 0.7354 | 0.0743 | 0.0973 | 0.0989 |
| Jukebox 72 (Teacher) | 0.3058 | 0.1985 | 0.3375 | 0.1143 |
| Jukebox 18 | 0.2109 | 0.2647 | 0.4162 | 0.0814 |
| Jukebox 18 (Student) | 0.3444 | 0.2165 | 0.2476 | 0.1292 |

## 6. Discussion

One of the main drawbacks of the Jukebox occurs when it is required to generate long samples. This is because it loses temporal traditional structure, which is essential in music as we perceive it. Music structures such as choruses that repeat or question–answer-like melodies cannot be generated with longer samplers. Some other token-based autoregressive models have dealt with this issue, such as AudioLM and MusicLM [30,39], even though in such cases, music and audio are conditioned on text and melody instead of lyrics, and they do not focus on sung music. In the future, experimenting with semantic tokens proposed by these two mentioned models with Jukebox transformer architecture could be the key to creating longer coherent music sequences. Additionally, using distillation to reduce conditional transformer parameters is in our future scope. Conditional training has the need of a paired data dataset, which is troublesome to obtain if it is based on copyrighted content. Making it possible to use unpaired data for training models conditioned on lyrics and metadata is also in our scope.

Another aspect is the categorical distribution modeling. The categorical non-continuous distribution is non-derivable, so it cannot be used in methods such as normalizing flows as in Parallel Wavenet [23]. Training transformer-like architectures for modeling discrete gaussians or logical distributions is currently a source of research. Finally, we acknowledge the progress in audio quality evaluation. Publicly available models, such as Trill [40] and VGGish [41], for audio quality metrics are expected to be tested with our model.

## 7. Conclusions

In this paper, the reach when distilling the transformer prior of the Jukebox model to accelerate audio generation was shown, which reduces the inference time of original Jukebox system. The proposed scheme generates audio signal several times faster than the smaller Jukebox architecture. Because the proposed system is intended to reduce only the inference time of the original Jukebox, only 300 songs by a single interpreter were used as training data. Evaluation results show that even though the audio quality was far from ideal, the proposed distilled version achieved a categorization more towards actively pleasant based on the emotional valence evaluation. The quality had some room for improvement, even though the sampling time was reduced. It is important to note that when sharing the same architecture as the baseline teacher, the student model was still autoregressive. A paralleling sampling procedure using non-autoregressive methods by distilling it with autoregressive ones is set to be explored in the future.

**Author Contributions:** Conceptualization, M.P.-M. and T.N.; methodology, T.N.; software, M.P.-M.; validation, M.P.-M., T.N. and H.P.-M.; formal analysis, H.P.-M.; investigation, M.P.-M.; resources, T.N.; data curation, M.P.-M.; writing—original draft preparation, M.P.-M.; writing—review and editing, H.P.-M.; visualization, M.P.-M.; supervision, H.P.-M. and T.N.; project administration,

H.P.-M.; funding acquisition, T.N. and H.P.-M. All authors have read and agreed to the published version of the manuscript.

**Funding:** This research was funded by the Instituto Politécnico Nacional of Mexico and the University of Electro-Communications of Japan.

**Institutional Review Board Statement:** Not applicable.

**Informed Consent Statement:** Not applicable.

**Data Availability Statement:** http://github.com/MichelPezzat/jukebox; http://soundcloud.com/michel-pezzat-615988723 (accessed on 22 February 2023).

**Acknowledgments:** The authors thank the Nakashika Laboratory of the University of Electro-Communications of Japan for support provided during the realization of this research.

**Conflicts of Interest:** The authors declare no conflict of interest.

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
