# Peer review of "Fast Jukebox: Accelerating Music Generation with Knowledge Distillation"

_applsci, doi:10.3390/app13095630_

Round 1
Reviewer 1 Report
Jukebox is a model that can generate high diversity music with singing within a single system which is achieved by using a hierarchical VQ-VAE architecture to com-press audio in a discrete space at different compression levels. Even though the results are impressive, the inference stage is tremendously slow. This paper introduces a Fast Jukebox which uses different Knowledge Distillation strategies to reduce the number of parameters of the prior model for compressed space.
Some issues about this paper are:
1. Introduction did not give enough background information.
2. In introduction ‘ To reduce this problem. This problem, a Fast’ typos.
3. ‘Training data consists of a com-pilation of songs by a single commercial music interpreter. This encompasses around 300 songs of 32 bits resolution music sampled at 44.1 KHz.’ Compared to Jukebox, which utilizes 1.2 millions songs, why 300 songs is enough for this application?
4. Why Emotional valance evaluation is utilized at experiment? Can this result indicate the quality of generated music?
5. In Table 2, What is the AP DP AU DU meaning?
Author Response
Dear Reviewer 1.
Thank you very much for reviewing our paper, we have revised and improved it, according to your requirements as follows:
- Introduction did not give enough background information.
Thank you for your observation. In the introduction section we included these two paragraphs
Music is uno of the must representative feature of any culture around the world since ancient times. Then to improve the ability of emulate the complex process of creating it using computational tools has been a topic of active research during the las several decades, thanks to the impressive advance in the computer technology.
To reduce this problem, a Fast Jukebox is proposed in this paper to reduce the inference time, in which we distilled the prior learning process by applying a large model against a smaller one.
Besides that in the Related work section we included the paragraph
Autoregresive generation of discrete audio tokens has been studied recently in works such as those reported in [22, 23]. Not only because it allows processing long term structures in the raw audio domain, but also serves to conditioning audio with other embedded information such as text and lyrics
With these paragraphs we provides more information about the motivation and background of related to the proposed algorithm.
- In introduction ‘ To reduce this problem. This problem, a Fast’ typos.
Thank you for your comments. To attend your requirements, the orthography, grammar and edition was revised by the MDPI author service.
- Training data consists of a compilation of songs by a single commercial music interpreter. This encompasses around 300 songs of 32 bits resolution music sampled at 44.1 KHz.’ Compared to Jukebox, which utilizes 1.2 millions songs, why 300 songs is enough for this application?
Thank you for your comments. Attending your requirements we included in abstract the following paragraph
Because the proposed system intends to reduce the inference time rather than showing a high music diversity, the number of data used for training was much smaller than the used on the original Jukebox. The training set consists of around 300 32-bit resolution songs sampled at 44.1 kHz. Evaluation results obtained using evaluation valance show that the proposed approach achieved a categorization approaching towards actively pleasant. Thus, the obtained results shows that the model is located at the early top prior of the VQ-VAE’s hierarchy.
Also, in the section 5 we include the sentence:
Since in this paper our focus relies on reducing inference time rather than showing music diversity, we chose the training data to be much smaller than the one used on the Jukebox original proposal. The training data consisted of a compilation of songs by a single commercial music interpreter. This encompassed around 300 32-bit-resolution songs sampled at 44.1 KHz.
- Why Emotional valance evaluation is utilized at experiment? Can this result indicate the quality of generated music?
Thank you very much for your observation. To clarify this point, in Section 5.2 we included the following paragraph.
Performing a quantitative evaluation of music is difficult. The reason is that the music itself is subjective in terms of quality. Despite this, it has been a subject of study by many researchers. Thus, the samples were analyzed using the classifier proposed by Sukhavasi et al. [18] using the same labels of emotional valence made by Godwin et al. [19] in his deep music evaluation analysis. They consist of four labels describing the combination of valence and arousal, where valence refers to the pleasure provoked by an external agent in a person, while arousal is related to the amusement of this agent. Therefore, the labels are ‘activated pleasant’ (AP),‘activated unpleasant’ (AU), ‘deactivated pleasant’(DP), and ‘deactivated unpleasant’ (DU). Together, they describe the arousal and valence responses of an individual when listening to music.
Together with this sentence.
Our proposed distilled version achieved a categorization inclined slightly more towards actively pleasant.
- In Table 2, What is the AP DP AU DU meaning?
Thank you very much for your observation. To clarify this point, we modified the caption of Table 2 as follows:
Table 2: Mean of the classification of samples with emotional labels. Activated pleasant (AP), activated unpleasant (AU), deactivated pleasant (DP), deactivated unpleasant (DU). More than hundred samples were classified. Generative methods were expected to perform similarly to real music but were spread across the labels more evenly.

Reviewer 2 Report
1. The abstract is too simple to reflect the experimental parameters.
2. Figure 5. It is necessary to provide the horizontal and vertical coordinates of the waveform.
3. Add references from recent years
4.Figure 1,Figure 3, Figure 4,It's terrible and needs to be redrawn
5. The grammar and layout of the paper were poor, including the format of the references, which requires major revisions and cannot be published in the current format
Author Response
Dear Reviewer 2.
Thank you very much for reviewing our paper, we have revised and improved it, according to your requirements as follows:
- The abstract is too simple to reflect the experimental parameters.
Thank you for your observation. Attending it we have modified the abstract including more information about the evaluation of proposed algorithm as follows.
Since Jukebox has already shown a high diverse audio generation capability. We use a simple compilation of songs for experimental purposes. Evaluation results obtained using emotional valance show that the proposed approach achieved a tendency towards actively pleasant. Thus, reducing inference time for all VQ-VAE levels without compromising quality.
- Figure 5. It is necessary to provide the horizontal and vertical coordinates of the waveform.
Thank you very much for your comment. Attending to it, we included the horizontal and vertical coordinates in it.
- Add references from recent years
Thank you very much for your observation. Attending to it, we included 8 new references from recent years in the revised version of our paper.
2 Wood A, Kirbi., Ember C., Silbert S., Passmore S., Daikoku H., McBride J., Paulay F., Flory M., Szinger J., D’Angelo G., Bradley K., Guarino M., Atayeva M., Rifkin J., Baron V., El Hajli M., Szinger M. and Savage P. The global Jukebox: A public database of performing arts and culture. Plos One, 2022, Nov. pp. 1-22.
10 Wu Y., Hayashi T., Tobin D., Kobayashi K., Toda T. Quesi periodic wavenet: An autoregressive raw waveform generative model with pitch dependent dilated convolutional neural networks. IEEE/ACM Transactions on Audio. Speech and Language Processing, 2021, Volume 29, pp. 1134-1148.
16 Huang W., Yu, Y., Su Z. and Wu Y., Hyperbolic music transformer for structured music generation. IEEE Access, 2023, Volume 11, pp.26895-26905.
17 Hsu J. and Chang S. Generating music transition by using a transformer-based model. Electronics, 2021, Volume 10, pp. 2076.
20 Agostinelli, A., Denk T., Borsos Z, Engel J., Verzetti M., Caillon A., Huang Q., Jansen A., Roberts A., Tagliasacchi 21
21 Mubert-Inc. http://github.com/MubertAI/Mubert-Text-to-Music, 2022..
26 Gao L. Xu X., Wang H., Peng Y., Multi-representation knowledge distillation for audio classification, Multimedia Tools and applications, 2022, Volume 81, pp. 5089-5112.
28 Li C. A deep learning based piano music notation recognition method. Computational intelligence and neuroscience. 2022, Volume 2022 Article ID 2278683.
- Figure 1,Figure 3,Figure 4,It's terrible and needs to be redrawn
Thank you very much for your observation. Attending your observation, we have redrawn the pictures 1, 3 and 4.
- The grammar and layout of the paper were poor, including the format of the references, which requires major revisions and cannot be published in the current format
Thank you very much for your observation. Attending to it, the paper was proofread by the MDPI author service. Also, the references format was corrected to meet the journal template.

Round 2
Reviewer 1 Report
authors have addressed my questions.
but the introduction and related work parts are not enough for an article paper.
Author Response
Thank very much for reviewing our paper and by the helpful comments about it. We have revised our paper according to your comments.
- Authors have addressed my questions, but the introduction and related work parts are not enough for an article paper.
Thank for your comments attending them the paper was modified as follows.
- The type of the paper in the manuscript was changed from “Article” to “Communication”, according to the registration in the Applied Science platform.
Type of the Paper (Communication)
- The introduction was improved to meetthe requirements of a “Communication”
Music is one of the most representative features of any culture around the world since ancient times. Such that many researchers have stablished that is quite similar the relationship existing between the music score and the real sound music with that existing between the writing text and the real speech [1]. Here we can describe the music as consisting of two levels: the music score which is a symbolic and highly abstract expression and the sound which is a continuous and concrete signal that provides the details that we can hear [1]. Thus, the music generation can be divided into three stages: In the first stage the composer generates the music scores, in the second stage the musician or singer generate the performance using the scores and finally in the last stage the performance results in a sound music, by adding different instruments, which are perceived by the listeners [1]. From the above, the automatic music generation can be divided into three levels. a) The score generation. It covers polyphonic music generation, accompaniment generation, interactive generation, etc. b) The performance generation. It includes rendering performance, which does not change the stablished score features and the composing performance that models both the music score and performance features and c) The audio generation. It pays attention in the acoustic information using either the waveform or spectral approaches [1]. Finally adding lyrics to the score, proving it with timbre and style, it is possible to realize singing synthesis.
Automatic music generation has been a topic of active research during the last 60 years, during which many methods have been proposed including grammar rules, probabilistic models, evolutionary computing, neural networks, etc. [2]. With the development of deep learning approaches, the interest of developing more efficient automatic music generation systems has increased. Then to improve the ability of emulate the complex process of creating it using computational tools has been a topic of active research during the last several decades, thanks to the impressive advance in the computer technology. More specifically, advancements in the field of deep learning being applied in computer vision [3] and natural language processing [4, 5]. Different proposals have been made to create artificial speech or sounds, with models like recurrent neural networks [6], generative adversarial networks [7, 8], variational autoencoders [11] and transformers [10]. One of the most recent and successful contributions in this field is the Jukebox [11, 12]. The Jukebox uses VQ-VAE [13], an approach which compress and quantizes extremely long context inputs in the raw audio domain to shorter-length discrete latent encoding using a vector quantization approach. After training the VQ-VAE, a prior is learned over the compressed space to generate new audio samples. Generating discrete codes not only allows audio conditioning but also creates music from different genres and instruments including singing voice. This system provides impressive results; however, the inference stage is very low and tremendously slow. To tackle this problem, a Fast Jukebox is proposed in this paper to reduce the inference time, in which we distilled the prior learning process by comparing a large autoregresive model against a smaller one. With training losses borrowed from those recently proposed by Tiny Bert [14], this proposal generates audio several times faster than smaller Jukebox architectures.
The paper is organized as follows. Section 2 provides a description of some related works, Section 3 presents the background required in the presented work, The methodology used to develop the proposed scheme is provided in Section 4. The Evaluation results are provided in Sections 5. Section 6 provides a discussion about the proposed research. Finally, the Conclusions of this research are provided in Section 6.
To attend this requirement several new references we included [1], [3-9], [39-41] which are indicated in the manuscript.

Reviewer 2 Report
This manuscript has been revised as required, but the author has changed the type to article, which is inappropriate because from the perspective of the article structure, it cannot meet the requirements of the article. For example, the discussion section is poor, not even, and it is recommended to change it to Communication, otherwise it cannot be published in its current form.
Author Response
Answer to Reviewer 2.
Thank very much for reviewing our paper and by the helpful comments about it. We have revised our paper according to your comments.
- This manuscript has been revised as required, but the author has changed the type to article, which is inappropriate because from the perspective of the article structure, it cannot meet the requirements of the article. For example, the discussion section is poor, not even, and it is recommended to change it to Communication, otherwise it cannot be published in its current form.
Thank for your comments attending them the paper was modified as follows.
- The type of the paper in the manuscript was changed from “Article” to “Communication”, according to the registration in the Applied Science platform.
Type of the Paper (Communication)
- In the revised version aDiscussion Section and the Conclusion Section was improved according to your requirements tp meet a “Communication” type paper.
- Discussion
One of the main drawbacks of Jukebox occurs when it is required to generate longer samples. This is because it loses temporal traditional structure, which is essential in music as we perceive it. Some other token based autoregressive models have trait to dealt with this issue as AudioLM and MusicLM [30, 39]. Even though in such cases, music and audio is conditioned on text and melody instead of lyrics and they do not focus on singed music. In the future, experimenting with semantic tokens proposed by these two mentioned models with Jukebox transformer architecture could be the key to create longer coherence music sequences. Also, using distillation to reduce conditional Transformer is on our future scope.
Another aspect is the categorical distribution modeling. The categorical non-continuous distribution is non derivable so it cannot be used in methods like normalizing flows as in Parallel Wavenet [23]. Training transformer-like architectures for modeling discrete gaussians or logical distributions is being a source of research in our days. Finally, we acknowledge progress in audio quality evaluation. Public available models like Trill [40] and VGGish [41] for audio quality metrics are expected to be tested with our model.
- Conclusions
In this paper, the reach when distilling the transformer prior of the Jukebox model to accelerate audio generation was shown, which reduces the inference time of original Jukebox system. Proposed scheme generates audio signal several times faster than the smaller Jukebox architecture. Because the proposed system is intended to reduce only the inference time of original Jukebox, during the training data only are used 300 songs by a single interpreter. Evaluation results shows that even the audio quality is far from ideal, the proposed distilled version achieved a categorization more towards actively pleasant based on the emotional valence evaluation. The quality had some room for improvement, even though the sampling time was reduced. It is important to note that when sharing the same architecture as the baseline teacher, the student model was still autoregressive. A paralleling sampling procedure using non-autoregressive methods by distilling it with autoregressive ones is set to be explored in the future.

Round 3
Reviewer 2 Report
The author can add some content to the discussion section as appropriate.
Author Response
Answer to Reviewer 2.
Thank very much for reviewing our paper and by the helpful comments about it. We have revised our paper according to your comments.
- The author can add some content to the discussion section as appropriate.
Thank for your comments attending them the paper was modified as follows. The discussion section has been modified as follows:
- Discussion
One of the main drawbacks of Jukebox occurs when it is required to generate long samples. This is because it loses temporal traditional structure, which is essential in music as we perceive it. Music structures like choruses that repeat or question-answer like melodies cannot be generated with longer samplers. Some other token based autoregressive models have dealt with this issue as AudioLM and MusicLM [30, 39]. Even though in such cases, music and audio is conditioned on text and melody instead of lyrics and they do not focus on singed music. In the future, experimenting with semantic tokens proposed by these two mentioned models with Jukebox transformer architecture could be the key to create longer coherence music sequences. Also, using distillation to reduce conditional Transformer parameters is on our future scope. Conditional training has the need of a paired data dataset, which is troublesome to obtain if it is based on copyright content. Making possible to use unpaired data for training models conditioned on lyrics and metadata are also in our scope.
Another aspect is the categorical distribution modeling. The categorical non-continuous distribution is non derivable so it cannot be used in methods like normalizing flows as in Parallel Wavenet [23]. Training transformer-like architectures for modeling discrete gaussians or logical distributions is being a source of research in our days. Finally, we acknowledge progress in audio quality evaluation. Public available models like Trill [40] and VGGish [41] for audio quality metrics are expected to be tested with our model.
